# Potential Diagnostic and Monitoring Biomarkers of Obstructive Sleep Apnea–Umbrella Review of Meta-Analyses

**DOI:** 10.3390/jcm12010060

**Published:** 2022-12-21

**Authors:** Piotr Fiedorczuk, Agnieszka Polecka, Marzena Walasek, Ewa Olszewska

**Affiliations:** 1Doctoral School of the Medical, University of Bialystok, 15-089 Bialystok, Poland; 2Department of Otolaryngology, Medical University of Bialystok, 15-276 Bialystok, Poland

**Keywords:** sleep apnea, biomarkers, intermittent hypoxia, inflammation, sleep disordered breathing

## Abstract

Obstructive sleep apnea (OSA) is a prevalent, underdiagnosed disease that imposes a significant impact on the health and wellbeing of patients and a financial burden on individuals, their families, and society. Development of new methods of testing other than an overnight sleep study, such as measurement of serum or plasma biomarkers, may provide an easier diagnostic process to identify patients with OSA and allow earlier initiation of treatment, which might prevent serious comorbidities. We conducted a systematic review and quality assessment of available meta-analyses regarding potential diagnostic and monitoring biomarkers of obstructive sleep apnea. A total of 14 sets of candidate biomarkers displayed differences in levels or concentrations in OSA patients compared to non-OSA controls, and decreased after OSA treatment: CRP, IL-6, TNF-α, Il-8, HCY, ICAM-1, VCAM-1, VEGF, TC, LDLc, HDLc, TG, leptin, MDA, ALT, AST, IGF-1, adiponectin, and cortisol. This review summarizes the evidence for OSA-associated potential biomarkers and demonstrates that the quality of available studies, as measured by AMSTAR2, is often low and associated with a high risk of bias.

## 1. Introduction

Obstructive sleep apnea (OSA) is a nocturnal disorder characterized by recurrent episodes of upper airway obstruction during sleep, associated with oxygen desaturation and sleep fragmentation [1,2,3,4,5]. It is an increasingly prevalent condition that greatly impacts public health. Epidemiologic data show it as a disorder with a high prevalence of 9% to 38% in the general adult population, from 13% to 33% in men, and from 6% to 19% in women [6].

Obstructive sleep apnea is responsible for the increased risks of cardiovascular and pulmonary illnesses, stroke, and operative and postoperative risks [7,8,9]. Patients with OSA often manifest daytime sleepiness and experience a high risk for workplace and traffic accidents [10]. There is a strong association between obstructive sleep apnea and traffic accidents. Apnea–hypopnea index (AHI) of greater than 15/hour is associated with multiple accidents in 5 years [11]. In patients with OSA, the risk of an accident is higher among those who had consumed alcohol on the day of the accident [12]. Patients with obstructive sleep apnea have significantly abnormal health-related quality of life (HRQoL) scores compared to healthy age- and gender-matched controls [13]. In summary, OSA has a significant impact on health and wellbeing and imposes a financial burden on individuals, their families, and society.

The pathogenesis of OSA is multifactorial and still not fully established. It involves a diverse range of mechanisms, including selective activation of inflammatory molecular pathways, endothelial dysfunction, metabolic dysregulation, and oxidative stress [14]. Endothelial dysfunction is often considered to be one of the earliest detectable and possibly reversible abnormalities during the development of atherosclerosis. Studies indicate an association between the presence of both coronary and systemic endothelial dysfunction and an increased risk for future cardiovascular morbidity and mortality in patients with OSA [15,16,17]. Consequently, the severity of endothelial dysfunction may depend on the severity of OSA. The development of cardiovascular morbidity and mortality may also occur secondary to other pathologies caused by OSA, such as hypertension and diabetes [18] (Figure 1).

Cardiovascular and pulmonary complications are thought to be caused by tissue injury associated with chronic intermittent hypoxia [19]. The mechanisms of these tissue injuries are not known. OSA stimulates, mainly through intermittent hypoxia, several mechanisms, including inflammation, endothelial dysfunction, and oxidative stress [20]. In intermittent hypoxia, the inflammatory signaling factors play essential roles in the transcriptional regulation of inflammatory cytokines. However, their role in OSA pathogenesis and the exact relationships between pathogenic mechanisms remain unclear.

Recurrent episodes of breathing cessation during sleep expose the cardiovascular system to cycles of significant hypoxia, exaggerated negative intrathoracic pressure, and arousals. In OSA cases, repetitive hypoxia and reoxygenation occur during sleep, inducing the oxidative stress response [20]. This oxidative stress not only damages endothelial cells in the peripheral vasculature but also contributes to the damage of alveolar epithelial cells [21]. Reduced alveolar trans-epithelial exchange rate for oxygen and carbon dioxide leads to worsened hypoxemia and hypercarbia. Increased permeability of the alveolar epithelium results in the development of intra-alveolar transudate and exudate. Moreover, the decreased level of pulmonary surfactants [22] leads to difficulty maintaining alveolar patency and preventing alveolar collapse and pulmonary atelectasis. Oxidative stress and chronic inflammatory state are the main characteristic pathophysiological changes in OSA contributing to neural, cardiovascular, and metabolic alterations. Thus, cardiovascular morbidity and mortality are increased in patients with OSA [23].

It is possible that OSA is associated with an increase or decrease of different compounds related to inflammation, oxidative stress, or changes in metabolism. 

C-reactive protein (CRP), a key blood inflammatory marker, is generated in the liver and is predominantly regulated by the proinflammatory cytokine interleukin-6 (IL-6). Unlike other cytokines, CRP levels are rather steady in the same individual over 24 h and may indicate the extent of the inflammatory response. Epidemiological research has demonstrated that an elevated CRP level in the high–normal range (0.2 to 1.5 mg/dL) in seemingly healthy men and women is a reliable indicator of cardiovascular risk. A higher CRP level is related to future cardiovascular events in patients with stable angina pectoris, acute coronary artery disease, and a history of myocardial infarction. CRP may directly contribute to the onset and progression of atherosclerosis. Endothelial cells, vascular smooth muscle cells, and monocyte macrophages contain proinflammatory and proatherogenic features, and CRP levels are also connected with oxidative stress [24].

It is generally recognized that the acute phase response is mediated by IL-6, which is a cytokine that possesses multiple functions. In addition to its role in host defense, inflammation, and cancer, IL-6 is also thought to contribute to the proliferation and hypertrophy of individual cells [25]. Increases in vascular cell-derived IL-6 have been connected with multiple stimuli, such as inflammatory cytokines and growth factors. These variables have been shown to cause an increase in IL-6 production. The levels of IL-6 are often elevated in cardiovascular diseases such as atherosclerosis and hypertension. It is believed that IL-6 is responsible for the structural and functional changes that occur in arteries as a result of these diseases.

Tumor necrosis factor-alpha (TNF-α) is a proinflammatory cytokine that plays a crucial role in host defense and mediates the pathogenesis of several disease processes, including atherosclerosis, septic shock, and autoimmune disease [26]. It is implicated in multiple necroses- and apoptosis-related signaling pathways. It also regulates sleep and has been associated with excessive daytime drowsiness, disturbed nighttime sleep, and hypoxia [27].

The chemoattractant cytokine interleukin-8 (IL-8) is produced by some different types of tissue and blood cells. IL-8 is largely released by mononuclear macrophages. When stimulated appropriately, epithelial and endothelial cells can also produce IL-8 [28]. In inflammatory areas, IL-8 attracts neutrophils and activates them. To maintain inflammation, IL-8 can cause neutrophils to produce myeloperoxidase and attract other inflammatory cells. IL-8 binds to certain receptors on the surface of neutrophils, causing cell deformation, degranulation, and an increase in reactive oxygen species production. This procedure could cause lysosomes to release their contents and activate arachidonic acid, increasing vascular permeability and plasma protein exudation in the process, which can result in tissue damage, atherosclerosis, vascular inflammation, and other disorders [29]. OSA patients may experience a rapid mobilization of macrophage antigen 1 to the neutrophil surface by exposure to IL-8. As a result, upregulation of IL-8 in human bronchial epithelial cells has been observed in response to a vibration stimulation brought on by snoring [30]. IL-8 influences physiological sleep in healthy individuals and is connected to physiological secretory patterns [31]. Decreased IL-8 secretion is linked to good sleep at night and a healthy physical state the following day, whereas high IL-8 secretion may be linked to excessive daytime sleepiness and weariness [32]. 

Intercellular adhesion molecule-1 (ICAM1) is a ligand for lymphocyte-function-associated antigen-1 and an 80 to 110 kDa glycoprotein with five immunoglobulin-like domains. According to reports, ICAM-1 is crucial for leukocyte migration to inflamed tissue [33]. The injurious factors associated with OSA (repetitive hypoxia, sympathetic nervous system activation, hypertension, obesity, insulin resistance, and dyslipidemia) induce the release of primary proinflammatory cytokines (e.g., interleukin-1 and tumor necrosis factor-alpha), and this stimulates the production of adhesion molecules, procoagulants, and other mediators by endothelial and other cells. Leukocyte adherence to the endothelium is mediated by intercellular adhesion molecule-1 (ICAM1) and vascular cell adhesion molecule-1 (VCAM-1). VCAM-1 is a part of the immunoglobulin superfamily of adhesion molecules and binds for very late antigen-4, which is found on monocytes and lymphocytes but not on neutrophils [34]. The earliest indication of disease activity in both animal and human models of atherosclerosis is an elevation of adhesion molecules [35]. Adhesion molecules mediate the attachment of circulating leukocytes to the endothelium, in addition to their subsequent transmigration and accumulation in the arterial intima. ICAM-1 in particular has been reported to play an important role in the transmigration of leukocytes across the vascular endothelial wall [33,35,36]. The expression of adhesion molecules is an indicator of endothelial inflammation and is likely to be involved in the causal pathway leading to atherosclerosis [36]. According to several studies, OSA-induced repeated hypoxia may play a role in the pathophysiology of cardiovascular diseases by inducing inflammatory responses through increasing cytokine and adhesion molecule levels [37,38]. ICAM-1 levels have been strongly linked to cardiovascular-disease-related deaths [39]. Additionally, they have been associated with cardiovascular disease risk factors and subclinical cardiovascular disease findings [40,41]. In older men and women, circulating ICAM-1 has been associated with cardiovascular disease risk factors and fatal events [42].

Vascular endothelial growth factor (VEGF), a soluble angiogenic mitogen, can promote angiogenesis and increase tissue capillary density [43]. It may have an impact on the prognosis of cancer [44], the process of atherogenesis [45], the development of cardiovascular disorders [46], and other conditions. Hypoxia can stimulate the high expression of the VEGF gene through the hypoxia-inducible factor (HIF) [47]. Experimental studies have shown that intermittent hypoxia, a defining feature of OSA, is related to elevated VEGF [48,49]. However, there has not been a clear consensus about the VEGF levels in OSA patients in human research.

Homocysteine (HCY) is a sulfur amino acid that is metabolized through two different pathways: transsulfuration to cystathionine and remethylation to methionine [50]. Disrupted homocysteine metabolism leads to hyperhomocysteinemia, a condition that has been linked in recent epidemiological studies to a higher risk of many diseases—increased HCY is a specific risk factor for cardiovascular diseases, dementia, and Alzheimer’s disease [51,52]. Hyperhomocysteinemia maintains oxidative stress, weakens endothelial dysfunction, and increases the metabolism of harmful cysteine adducts [53]. A proportional risk of mortality is independently correlated with blood HCY levels [54]. Increased reactive oxygen species (ROS) could be linked to an inflammatory reaction occurring in OSA. Additionally, the local inflammatory reaction brought on by repeated upper airway collapse could potentially result in a systemic inflammatory reaction. Homocysteine has a highly reactive sulfhydryl group. The sulfhydryl group is readily oxidized by itself [55]. More than 98% of homocysteine is in an oxidized condition [56]. The vascular endothelium can sustain direct harm from activated ROS and white blood cells. The ability of oxidative stress in vivo diminishes as oxidative stress products increase in OSA patients [57]. Therefore, homocysteine is crucial in the development of oxidative stress in OSA patients.

Malondialdehyde (MDA) is an end-product of enzymatic or nonenzymatic decomposition of arachidonic acid and larger PUFAs [58]. During the formation of thromboxane A2, enzymatic pathways can produce MDA in vivo as a byproduct. Once generated, MDA can be processed by enzymes or react with DNA or proteins in cells and tissues to form adducts, which can cause biomolecular damage. The primary basis for MDA’s high reactivity is its electrophilicity, which renders it highly reactive toward nucleophiles such as basic amino acid residues (i.e., lysine, histidine, or arginine). Because MDA adducts can promote intramolecular or intermolecular protein/DNA crosslinking, which may cause significant alteration in the biochemical properties of biomolecules and accumulate during aging and in chronic diseases, they play an important role in secondary harmful reactions (such as cross-linking) [59,60,61]. Low-density lipoproteins, elastin, and collagen are examples of molecules that can be altered by MDA and may have effects on cardiovascular disease. The duration of nocturnal desaturation below 85% is linked with MDA concentrations. MDA plays a crucial role in the analysis of oxidative stress measurements in OSA because it is a significant component of thiobarbituric acid reactive chemicals [62,63].

The main source of circulating leptin is white adipose tissue. The main effects of leptin are to indicate satiety and to decrease the need to eat. Leptin and the leptin receptor are also implicated in the control of immunological function, inflammation, blood glucose metabolism, and energy expenditure [64]. When a person is obese, leptin levels in the blood are markedly elevated. The most important risk factor for the development of OSA is obesity. 

IGF-1 is a representative of the peptide hormone family. Its structural makeup is remarkably similar to that of proinsulin. As a mitogenic and metabolic factor, it significantly affects cell growth and metabolism. IGF-1 is primarily produced in the liver [65]. Sex steroids are the primary regulators of local IGF-I synthesis in the reproductive system, while growth hormone, parathyroid hormone, and sex steroids control IGF-I production in bone. IGF-I has a major function in controlling growth after birth. At birth, the concentrations are modest; they rise significantly during childhood and puberty, then, starting in the third decade, they start to fall [66]. IGF-1 deficiency can cause significant problems in healthy growth. IGF-1 level is downregulated in T1DM, whereas in T2DM, it is upregulated. IGF-1 elevation in adulthood may be associated with a higher chance of developing cancer. Moreover, treating IGF-1 in sepsis can offer therapeutic protection [67].

Alanine aminotransferase (ALT) and aspartate aminotransferase (AST) are liver transaminases aggregated in the cytosol of hepatocytes. In the serum, these enzymes are typically detectable at low concentrations of around 30 IU/L [68,69,70]. However, AST and ALT are released in greater quantities into the serum as a result of any procedure that results in the loss of hepatocyte membrane integrity or necrosis [71]. Most OSA patients are overweight, which places them at risk for fatty liver [5,72]. Numerous cases of ischemic hepatitis in severe OSA patients have been reported, and epidemiological studies have revealed that OSA is an independent risk factor for impairment of glucose homeostasis [73,74,75,76]. These data all point to the possibility that OSA, per se, could be a risk factor for liver injury independent of obesity. Therefore, direct liver hypoxia and OSA-induced insulin resistance may play a role in the development of liver disease linked to OSA.

Total cholesterol (TC), low-density lipoprotein cholesterol (LDL), high-density lipoprotein cholesterol (HDL), and triglyceride (TG) are components of the lipid profile. The risk of coronary heart disease and LDL are strongly positively correlated. The risk of cardiac mortality, nonfatal myocardial infarction, ischemic stroke, and the requirement for revascularization treatments is decreased when LDL cholesterol is lowered with medication, according to randomized studies. Although a direct link between OSA and dyslipidemia has not yet been established, there is mounting evidence that chronic intermittent hypoxia, a major OSA component, is independently related to and may even be the underlying cause of dyslipidemia due to the production of stearoyl-coenzyme A desaturase-1 and reactive oxygen species, lipid peroxidation, and dysfunction of the sympathetic nervous system. Alterations in oxidative stress and immune–inflammatory response are promoted by intermittent hypoxia associated with sleep apnea. In comparison to controls, OSA patients exhibit greater levels of systemic inflammatory markers. Human vascular endothelial cells’ LDL metabolism may be altered by cytokines, particularly IL-1, which may also affect how cholesterol is metabolized by endothelial cells. These alterations in the metabolism of endothelial cells offer proof of the pivotal function of cytokines in atherogenesis and other comorbidities.

The biomarker is referred to as a specified property that is assessed as an indication of normal biological processes, pathogenic processes, or reactions to an exposure or intervention. This wide definition includes various measurements, which might be obtained from genetic, histologic, radiographic, or physiologic properties [77]. 

A **diagnostic biomarker** detects or verifies the existence of an illness or condition of interest, or identifies a person who has a certain disease subtype, such as sweat chloride, which is utilized as a diagnostic biomarker to confirm cystic fibrosis [78]. 

A biomarker is considered a **monitoring biomarker** when it can be evaluated repeatedly to determine the progression of a disease or medical condition, to look for signs of exposure to a medical product or environmental agent, or to identify a medical product’s or biological agent’s effects [77]. For example, the international normalized ratio (INR) or prothrombin time (PT) may be utilized in order to determine if the intended impact of anticoagulation has been achieved in warfarin-taking individuals [79].

The evaluation of a biomarker that can be tested with appropriate accuracy and reliability in a defined context of application remains difficult. The objective is to develop a validation procedure that ensures the biomarker can be tested reliably, accurately, and consistently at a low cost.

Disease specificity, obligatory presence in all affected patients (i.e., high sensitivity and specificity), reversibility upon correct therapy, and detectability before patients exhibit visible clinical signs are all important aspects of the perfect biomarker. In addition, ideal biomarkers should not only represent the severity of the disease but also provide insightful data over the disease’s cumulative history, as well as allow for a cut-off value with minimum overlap between normal and disease. Furthermore, an ideal diagnostic policy based on biomarkers would be expected to reduce the total cost and burden of diagnosing a patient, measurement costs, and misdiagnosis costs. 

The standard diagnostic procedure for establishing the presence of obstructive sleep apnea (OSA) is overnight polysomnography (PSG). More than five scoreable respiratory events (e.g., apneas, hypopneas, RERAs) per hour of sleep need to be detected on PSG, as well as evidence of breathing effort for all or part of each respiratory event. Recent Clinical Practice Guidelines for OSA expressed the need for a new clinical screening or diagnostic tool for establishing the presence and severity of OSA [80]. More accurate and user-friendly screening and monitoring tools, such as serum and plasma biomarkers, may someday be the way to better diagnose OSA and assess its severity. 

New methods of testing other than an overnight sleep study may provide an easier diagnostic process to identify patients with OSA and allow earlier dispense of treatment, e.g., positive airway pressure (PAP) therapy, thus preventing serious comorbidities. Additionally, a monitoring tool for treatment evaluation and possible disease complications would be an important asset for clinicians. 

### Aim of the Study

We conducted a systematic review of available meta-analyses (2a level of evidence according to Centre for Evidence-Based Medicine, Oxford) [81]. The aim is to clarify which potential biomarkers of OSA have diagnostic and monitoring potential and show different concentrations between OSA subjects and non-OSA controls in addition to a change after OSA treatment.

## 2. Methods

The criteria of the Preferred Reporting Items for Systematic reviews and Meta-Analysis (PRISMA) checklist were followed in conducting and reporting this systematic review of meta-analyses [82]. Our PICO (population, indicator, control, outcome) question is shown in Table 1.

We searched PubMed and Scopus Library for meta-analysis articles concerning sleep apnea diagnostic and monitoring biomarkers. The search was performed using the words “sleep apnea”, “disordered breathing”, “biomarkers”, and “meta-analysis” in different combinations.

We searched the PubMed database using the following string: (sleep apnea) OR (OSAS) OR (disordered breathing) Filters: Meta-Analysis.

To obtain literature from the Scopus library, we used the following string:

TITLE-ABS-KEY (sleep AND apnea AND biomarker) OR (disordered AND breathing AND biomarker) AND (LIMIT-TO (SUBJAREA, “MEDI”)) AND (LIMIT-TO (EXACTKEYWORD, “Meta Analysis”) OR LIMIT-TO (EXACTKEYWORD, “Meta-Analysis”)). 

Search results were exported to the Mendeley reference manager for the initial title and abstract screening of the records. Duplicate articles were removed by the “remove duplicates” function of Mendeley. The literature search was performed between 10 June 2022 and 21 June 2022 and again on 10 September 2022. To obtain articles not received from databases, bibliographies of published articles were manually reviewed to identify additional studies. Two authors independently performed the literature search and evaluated articles for inclusion. Discrepancies, if any, were resolved with discussion.

During the initial screening of titles and abstracts, the studies retrieved had to meet the following criteria for inclusion in full-text eligibility assessment: (1) meta-analysis papers; (2) papers concerning adult human subjects with OSA; (3) measurement of serum or plasma compound. Due to language limitations, only articles written in English were selected. Papers evaluating a solely pediatric population (i.e., age < 18 years) were excluded from the search. Before excluding a potential biomarker based on a lack of corresponding studies regarding its diagnostic or monitoring capabilities, a manual database search was conducted to ensure there were no unfounded exclusions. After the initial screening, full-text manuscripts were retrieved and independently assessed by two investigators. The process for selecting the studies is provided in the flow chart in Figure 2. 

A quality score evaluation for studies was performed with AMSTAR2 (A MeaSurement Tool to Assess systematic Reviews) [83]. A total of 16 items were scored independently by 2 reviewers, and conflicting assessments were resolved by consensus. Items are listed in the Appendix A.

We selected a total of 50 meta-analyses of 14 potential sets of biomarkers (Table 2). ICAM-1, VCAM-1, ALT, and AST and lipid profile (TC, LDLc, HDLc, TG) were markers investigated in studies together, with similar results—in this review, they are described in conjunction.

### 2.1. Potential Biomarkers of OSA in Adults

#### 2.1.1. CRP

CRP was recognized to be elevated in OSA patients compared to healthy controls as early as 2002 [84]. Shamsuzzaman et al. found that in 22 OSA patients, compared to 20 control subjects, CRP levels were independently associated with OSA severity. Since then, many studies investigated the relationship between CRP/high-sensitivity CRP (hs-CRP) in OSA patients compared to healthy control subjects, and significant decreases in CRP levels after CPAP treatment in OSA patients were reported. A summary of studies regarding CRP chosen in this review is shown in Table 3.

Numerous studies have shown that CRP/hs-CRP levels are independently related to OSA. However, it is unknown whether there is a connection between the severity of OSA and elevated CRP or hs-CRP levels. Additionally, there was a strong correlation between OSA treatment, both CPAP therapy and sleep surgery, and CRP decrease. 

According to a study by Yi et al., both CRP and TNF-α were shown to be elevated in OSA, and the severity of the condition affected these rising tendencies. Effective CPAP therapy can, albeit gradually, lower increased CRP and TNF-α levels. Mendelian randomization analysis also found a possible causal link between OSA and increased CRP. This recent discovery further cements CRP as a possible diagnostic marker for OSA.

#### 2.1.2. Interleukin-6

In their 2015 scoping review, de Luca Canto et al. [97], after analyzing 117 biomarker studies in adults, concluded that IL-6 has a promising profile for screening and diagnosing patients with obstructive sleep apnea (OSA) syndrome. The summary of meta-analysis evidence of the IL-6 role in OSA is shown in Table 4. 

#### 2.1.3. Tumor Necrosis Factor-Alpha

Blood monocytes from OSA patients have been shown to produce large quantities of TNF-α. In addition, various studies have demonstrated that these patients had greater levels of TNF-α in the morning and shortly after the onset of obstructive apnea. Most studies indicated that patients with OSA have higher concentrations of serum and plasma TNF-α. The concentrations of TNF-α are decreased after PAP therapy treatment and sleep surgery. The summary of meta-analysis evidence of the TNF-α role in OSA is shown in Table 5.

#### 2.1.4. Interleukin-8

The concentrations of IL-8 in the serum and plasma are higher in subjects with OSA, compared to the control group, and decrease after PAP therapy treatment. The summary of meta-analysis evidence of the IL-8 role in OSA is shown in Table 6.

#### 2.1.5. ICAM-1 and VCAM-1

ICAM-1 and VCAM-1 were two of the first potential biomarkers assessed by de Luca Canto et al. [97]. Subjects with OSA exhibit higher serum concentrations of ICAM-1 and VCAM-1, which are reduced after treatment. The summary of meta-analysis evidence of the role of ICAM-1 and VCAM-1 in OSA is shown in Table 7.

#### 2.1.6. VEGF

According to several studies, VEGF is higher in OSA patients than in healthy controls, and OSA is associated with higher VEGF regardless of confounding variables [102,103]; meanwhile, additional research did not demonstrate comparable results in the OSA population [104,105]. The concentrations of VEGF decrease after PAP therapy treatment. The summary of meta-analysis evidence of VEGF’s role in OSA is shown in Table 8.

#### 2.1.7. Homocysteine

Studies have found higher concentrations of serum HCY in OSA subject in comparison to healthy controls. CPAP reduces HCY concentrations in OSA subjects. The summary of meta-analysis evidence of HCY’s role in OSA is shown in Table 9.

#### 2.1.8. Malondialdehyde

The concentrations of MDA in serum are higher in OSA patients compared to non-OSA controls. CPAP lowers serum and plasma concentrations of MDA in OSA patients. The summary of meta-analysis evidence of MDA’s role in OSA is shown in Table 10.

#### 2.1.9. Leptin

Subjects with OSA exhibit higher leptin concentrations in plasma. CPAP reduces leptin levels in OSA patients. The summary of meta-analysis evidence of leptin’s role in OSA is shown in Table 11.

#### 2.1.10. IGF-1

In comparison to healthy persons, OSA patients have considerably lower plasma/serum IGF-1 concentrations. Serum IGF-1 levels in patients with OSA may be influenced by the degree of the disease and ethnic variances. Additionally, the levels of plasma/serum IGF-1 correlated with minimal oxygen saturation and negatively correlated with the apnea–hypopnea index and oxygen desaturation index scores. This relation is independent of factors such as age, sample detection method, or study design [120].

Chen et al. suggested in their meta-analysis that in OSA patients, CPAP was linked to a statistically significant rise in IGF-1 [120]. Results of appropriate OSA studies regarding IGF-1 levels in comparison to healthy controls and change of IGF-1 levels after CPAP treatment are shown in Table 12.

#### 2.1.11. ALT and AST

The concentrations of ALT and AST are higher in subjects with OSA compared to the control group and decrease after PAP therapy treatment. The summary of meta-analysis evidence of ALT’s and AST’s roles in OSA is shown in Table 13.

#### 2.1.12. Lipid Profile

Patients with OSA have increased dyslipidemia compared to non-OSA controls. AHI had a substantial impact on LDL and TG, and BMI had a large impact on LDL and HDL concentration. CPAP and sleep surgery improved lipid profile. The summary of meta-analysis evidence of lipid profile’s role in OSA is shown in Table 14.

#### 2.1.13. Adiponectin

The concentrations of adiponectin in serum and plasma are lower in OSA patients compared to non-OSA controls. CPAP did not change the adiponectin levels in OSA patients after the therapy. The summary of meta-analysis evidence of adiponectin’s role in OSA is shown in Table 15.

#### 2.1.14. Cortisol

Although PAP treatment decreased plasma levels of cortisol in OSA subjects, available meta-analyses found that cortisol serum and plasma concentrations did not differ between OSA subjects and non-OSA controls. The summary of meta-analysis evidence of cortisol’s role in OSA is shown in Table 16.

### 2.2. Quality of Studies

We conducted a quality assessment of the studies using the AMSTAR2 tool. As written by the authors in the AMSTAR2 guidance document, AMSTAR2 is not intended to produce a final “score”. A high score could conceal significant flaws in a particular area, such as a poor literature search or a failure to consider the risk of bias (ROB) for each study that was included in a systematic review or meta-analysis.

No meta-analysis received the full score. The most common flaws were associated with the explanation and prior publication of the study protocol, insufficient study design, no justification of excluded studies, or incomplete individual studies ROB assessment. The final selection of studies with AMSTAR2 quality scores is shown in the Appendix A.

## 3. Discussion

Given its rising incidence and effects on the healthcare system, economy, and society, OSA has come to be recognized as a serious public health problem on a global scale. However, existing diagnostic techniques have shortcomings, which makes OSA clinical management difficult. Alternative approaches, such as biomarkers, are needed to direct medical decision-making. We aimed to gather possible OSA biomarkers for diagnostic and prognostic reasons in this systematic study.

Our review is an attempt to encapsulate the state of knowledge regarding probable new ways of OSA screening, diagnosing, monitoring, and prognosis. We summarized and evaluated the quality of available meta-analyses regarding potential diagnostic and monitoring biomarkers in OSA patients.

We included meta-analyses that inadvertently shared subject groups with other studies that were included, causing study overlap. There is a limited number of published studies, so performing a systematic review and meta-analysis of all the studies regarding a specific potential biomarker will result in investigating the patient group previously researched.

This is an umbrella review of the existing meta-analyses, and it does not combine data from the previously published meta-analyses or perform a new statistical analysis using the combined meta-analysis data. If possible, such work would require the identification of the individual studies that were included in each meta-analysis. To avoid over-representation of those studies, they should be removed after being included for the first time. Such work would require performing another meta-analysis, reaching, perhaps, a different conclusion, due to including a different set of original studies, but it would not be an umbrella review of the meta-analyses, as we conducted in this study.

In regards to the diagnostic capability of various biomarkers for OSA, most included studies reported differences between OSA subjects and healthy controls. We found only two meta-analyses comparing the sensitivity and specificity of using biomarkers to diagnose OSA. De Luca Canto et al. set out to compare the diagnostic significance of biological markers (exhaled breath condensate, blood, salivary, and urine) to the gold standard of nocturnal PSG in the diagnosis of OSA. They analyzed nine studies, four of which involved children and five of which involved adults, and concluded that only kallikrein-1, uromodulin, urocortin-3, and orosomucoid-1, when examined together, had sufficient accuracy to be an OSA diagnostic test in children. In adults, plasma levels of IL-6 and IL-10 may be useful indicators for determining whether OSA is present.

A follow-up systematic review by Gaspar et al. analyzed data from 16 studies with a total of 2156 individuals, of which 1369 had OSA diagnoses and 787 were healthy controls. Only two of the 38 biomarker candidates analyzed were examined in multiple studies. The most promising candidates for OSA diagnosis were identified by the mRNA levels of ADAM29, FLRT2, and SLC18A3 in PBMCs, Endocan, and YKL-40 in serum, and IL-6, and Vimentin in plasma.

Gaspar et al. found various limitations in current OSA diagnostic biomarker research. Many studies did not include clinical factors such as PSG results or clinical history and demographic data such as race/ethnicity. The majority of the included studies were single-center, had small sample sizes, and had a wide range of assessed/reported clinical and demographic characteristics. The majority of this research was either based solely or primarily on male cohorts or did not particularly address sex differences. They emphasized the urgent need for adopting better practice guidelines, for reporting to be improved, and for procedures to be standardized to increase the reproducibility and comparability of investigations.

The approach for OSA treatment is constantly developing. The golden standard is PAP therapy, the effect of which varies regarding patient compliance. There is also growing frequency and usage of soft-tissue surgical procedures for OSA, particularly in those with moderate to severe OSA who cannot use PAP therapy. Other forms of treatment, such as oral appliances or hypoglossal nerve stimulation therapy, are also increasingly common. For this review, we included meta-analyses that measured plasma and serum biomarkers after CPAP treatment and sleep surgery, due to the lack of research regarding other forms of treatment.

Although thorough, this review of meta-analyses has its limitations. Firstly, for the potential biomarker to be included, there must have been a meta-analysis study for comparison between OSA patients and the non-OSA control group, including regarding its change after treatment. This proves that many potential diagnostic and monitoring biomarkers are still in the early stage of research.

Moreover, meta-analyses included in this review are of varying quality, as scored in the AMSTAR2 assessment. More recent studies scored higher on average, possibly due to more frequent planning and preregistering protocols for systematic reviews and meta-analyses and tools for risk of bias assessment, such as the Newcastle–Ottawa scale or Cochrane ROBINS-I instrument.

Lastly, even though we tried to mitigate the risk of bias in our review regarding data search, selection and extraction, and quality assessment in our study by various means, there still may be a degree of bias. Studies might have been missed due to inadequate search strategy and not researching other databases during abstract screening or full-text reading.

## 4. Conclusions

There is a moderate level of evidence that OSA is associated with increased levels of serum and plasma inflammatory cytokines, oxidative stress indicators, adhesion molecules, adipose tissue hormones, abnormal lipid profile, and elevated liver enzymes, which can be decreased with CPAP treatment, namely, CRP, IL-6, TNF-α, Il-8, HCY, ICAM-1, VCAM-1, VEGF, TC, LDLc, HDLc, TG, leptin, adiponectin, cortisol, MDA, ALT, AST, and IGF-1. Individual potential biomarkers are reduced after sleep surgery treatment. Further, low-bias, high-quality studies and randomized control trials and meta-analyses of such are required to develop procedures that utilize serum or plasma biomarkers for diagnostic, prognostic, or monitoring purposes.

## Figures and Tables

**Figure 1 jcm-12-00060-f001:**
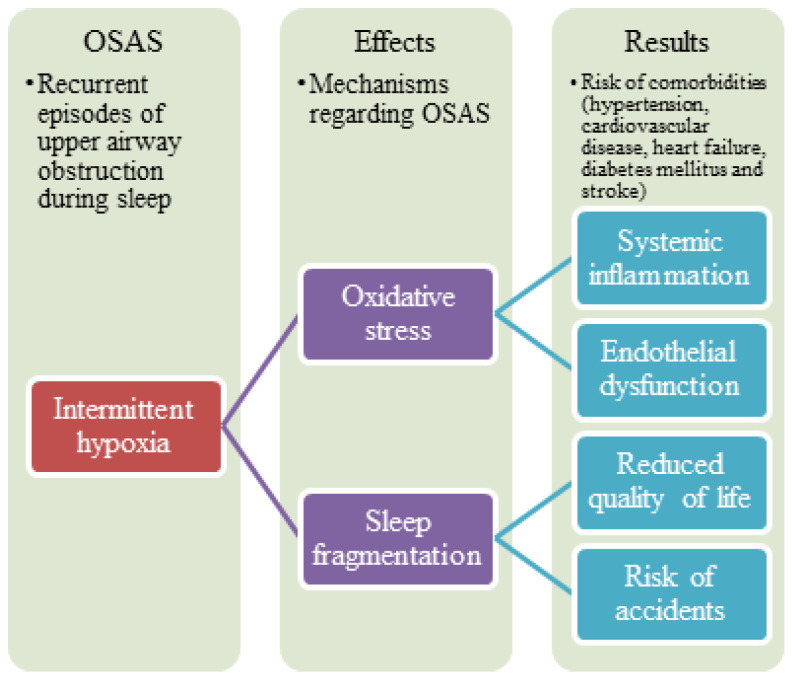
Summary of mechanisms regarding obstructive sleep apnea syndrome.

**Figure 2 jcm-12-00060-f002:**
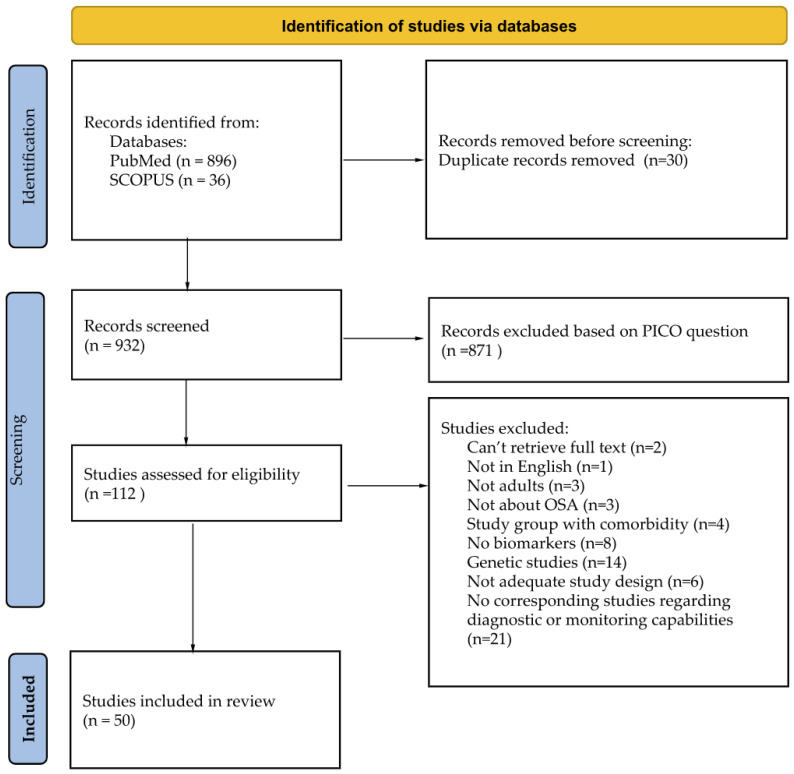
Studies search and inclusion flow chart.

**Table 1 jcm-12-00060-t001:** PICO(s) question.

Can Patients with Obstructive Sleep Apnea be Diagnosed and Monitored During Treatment Using Serum or Plasma Biomarkers?	
**The population**	Patients with OSA defined as AHI > 5 in polysomnographic sleep study (PSG).
**The indicator**	Serum or plasma biomarkers.
**The control**	Groups of patients without OSA (AHI < 5 in PSG), other OSA patients, same patients after treatment.
**The outcome**	The difference in serum or plasma biomarker levels.
**The study design**	Peer-reviewed English articles. Adult (>18 years) human subjects. Meta-analysis of case–control/observational studies, randomized control trials. Meta-analyses regarding differences in serum or plasma potential biomarker levels between obstructive sleep apnea subjects and controls. OR Meta-analyses regarding changes in serum or plasma potential biomarker levels in obstructive sleep apnea subjects after treatment.

Abbreviations: OSA, obstructive sleep apnea, AHI, apnea–hypopnea index.

**Table 2 jcm-12-00060-t002:** Number of chosen meta-analyses regarding individual potential biomarkers.

Potential Serum or Plasma Biomarker	The Number of Meta-Analyses Selected	
	Comparison to the Control Group	Change after Treatment
CRP	5	9
IL-6	6	6
TNF-α	5	7
Il-8	3	1
HCY	2	1
ICAM-1, VCAM-1	2	2
VEGF	2	1
TC, LDLc, HDLc, TG	1	4
Leptin	1	3
Adiponectin	1	3
MDA	1	2
ALT, AST	1	1
IGF-1	1	1
Cortisol	1	1

Abbreviations: CRP, C-reactive protein; IL-6, interleukin-6; TNF-α, tumor necrosis factor-alpha; IL-8, interleukin-8; Hcy, homocysteine; ICAM-1, intercellular adhesion molecule 1; VCAM-1, vascular cell adhesion molecule 1; VEGF, vascular endothelial growth factor; TC, total cholesterol; LDLc, low-density lipoprotein cholesterol; HDLc, high-density lipoprotein cholesterol; TG, triglyceride; MDA, malondialdehyde; ALT, alanine transaminase; AST, aspartate aminotransferase; IGF-1, insulin-like growth factor.

**Table 3 jcm-12-00060-t003:** Meta-analyses of CRP studies in OSA adults.

Authors	Material	Results and Conclusion
Comparison to the Control Group		
Imani et al., 2021 [85]	96 studies: 11 studies, 2994 subjects (plasma hs-CRP); 44 studies, 5097 subjects (serum hs-CRP); 9 studies, 938 subjects (plasma CRP); 32 studies, 3877 subjects (serum CRP)	Adults with OSA had considerably higher plasma and serum levels of hs-CRP and CRP than controls, although there was no discernible difference in plasma CRP levels between adults with OSA and controls. Plasma hs-CRP levels pooled MD in adults with OSA—0.11 mg/dL (*p* < 0.00001). Serum hs-CRP levels with OSA in adult 0.09 mg/dL (*p* < 0.00001). Plasma CRP pooled MD 0.06 mg/dL (*p* = 0.72). Compared to controls, the pooled MD of serum CRP levels in adults with OSA was 0.36 mg/dL (*p* < 0.00001).
Van der Touw et al., 2019 [86]	5 studies, 335 subjects	CRP levels in nonsmoking OSA participants without comorbidities were increased relative to levels in healthy matched nonsmoking control participants. SMD was 0.61mg/dL higher in OSA participants than in control group (CI 0.38–0.84, *p* < 0.00001).
Li et al., 2016 [87]	15 studies, 1297 subjects	Serum CRP/hs-CRP levels higher in moderate–severe OSA patients compared with control subjects. Serum CRP levels in the OSA group were 1.98 mmol/L higher than those in control group (95% CI: 1.39–2.58, *p* < 0.01). Serum hs-CRP levels in the OSA group were 1.57 mmol/L higher than those in the control group (95% CI: 0.96–2.18, *p* < 0.01).
Wang et al., 2015 [8]	18 studies, 1160 subjects	Serum levels of high-sensitivity C-reactive protein and C-reactive protein were significantly higher in patients with OSA than in controls (SMD 0.58, 95% CI 0.42–0.73, *p* < 0.0001).
Nadeem et al., 2013 [30]	30 studies, 4283 subjects	Patients with OSA had a statistically significant higher level of CRP when compared to control individuals (SMD, 1.77).
**Change after treatment**		
Yeo et al., 2022 [88]	11 studies, 397 subjects	Evidence suggested that individuals with OSA who underwent soft-tissue sleep surgery saw a reduction in CRP; this reduction was significant (SMD, 0.377; 95% CI, 0.617 to 0.137).
Yi et al., 2022 [89]	64 studies, 4285 OSA patients, and 3692 controls	Following CPAP intervention, CRP was marginally lower in the CPAP group compared to the non-CPAP group (WMD (95% CI) = 0.91 (1.65, 0.17) mg/L, *p* = 0.02).
Wang et al., 2022 [90]	34 studies, 1385 patients	Subgroup analysis showed that short-term CPAP decreases CRP (SMD: 0.73, 95% CI: 0.15–1.31; *p* = 0.011).
Kang et al., 2022 [91]	9 studies, 235 adults	Adult CRP levels significantly dropped after sleep surgery for OSA. Patients with OSA who underwent sleep surgery saw a considerable decline in their CRP levels (standardized mean difference (SMD) = −0.39, 95% CI, −0.67 to −0.11). Patients who have seen significant OSA improvement following sleep surgery (i.e., an AHI reduction of >20 events/h) experience a stronger positive effect of surgery on CRP levels.
Ning et al., 2018 [92]	15 studies, 1090 patients	CPAP reduces the inflammatory marker hs-CRP in OSA patients. The serum hs-CRP levels may serve as a predictor of how well OSA patients respond to CPAP therapy (SMD 0.64; 95% (CI) −1.19 to −0.09; *p*= 0.02).
Guo et al., 2013 [93]	14 studies, 1199 patients	In individuals with OSA, elevated CRP was found and was considerably decreased by successful CPAP treatment. Clinically, the utilization of CRP levels may be acknowledged as a useful predictor for OSA therapy monitoring (SMD 0.64; 95% (CI) 0.40 to 0.88; *p* = 0.000).
Baessler et al., 2013 [94]	14 studies with 771 patients	The serum inflammatory marker CRP is considerably reduced in OSA patients who use CPAP (pooled MD 0.14; 95% (CI) 0.08 to 0.20, *p* < 0.00001).
Xie et al., 2013 [95]	24 studies, 1597 patients	The combined data demonstrated that OSA patients’ CRP levels might be reduced by CPAP treatment (SMD 0.452; 95% (CI), 0.252–0.651).
Friedman et al., 2012 [96]	10 studies, 325 patients	Patients with OSA treated with CPAP seem to have a considerable reduction in CRP levels.

Abbreviations: OSA, obstructive sleep apnea; hsCRP, high-sensitivity C-reactive protein; CPAP, continuous positive airway pressure; CRP, C-reactive protein.

**Table 4 jcm-12-00060-t004:** Meta-analyses of IL-6 studies in OSA adults.

Authors	Material	Conclusion
Comparison to the control group		
Gaspar et al., 2022 [98]	16 studies, 1369 cases, 787 controls	IL-6 levels in plasma revealed the most promising candidates to further explore in future studies, as single or clustered biomarkers.
Yi et al., 2022 [99]	48 articles, 1974 patients, 1657 controls	OSA patients had greater IL-6 levels than controls—the weighted mean difference was 3.89. In comparison to the control, there were substantial differences for IL-6 with severe OSA severity; the causal relationship between IL-6 levels and OSA was insignificant, with an OR of 0.853 and *p* = 0.114.
Imani et al., 2020 [100]	63 studies: 39 studies, 2558 patients, 1897 controls (serum IL-6); 22 studies, 974 patients, 692 controls (plasma IL-6);	Higher levels of IL-6 have been linked to OSA severity. There were no significant independent effects of the publication year, mean BMI, mean AHI, or the number of participants on serum (pooled MD 2.89 pg/mL; *p* < 0.00001) or plasma IL-6 levels (pooled MD 2.89 pg/mL, *p* < 0.00001).
Zhong et al., 2015 [101]	31 studies, 1666 cases, 989 controls	Levels of IL-6 are higher in patients with OSA and are significantly correlated with OSA severity. IL-6 levels were higher in OSA compared to controls (SMD = 1.56, 95% CI = 1.18–1.95).
Nadeem et al., 2013 [30]	19 studies, 1316 subjects	IL-6 levels were greater in OSA patients than in healthy controls. A significant association was found between IL-6 and AHI. SMD 2.16.
De Luca Canto et al., 2015 [97]	1 study, 88 cases, 32 controls	Exhaled breath condensate IL-6 has the potential to predict the severity of OSA in nonsmoker OSA suspects.
**Change after treatment**		
Lee et al., 2022 [102]	6 studies, 140 subjects	Sleep surgery lowers IL-6 levels in adults with OSA (MD of 0.6 pg/mL and SMD of 0.66).
Yeo et al., 2022 [88]	5 studies, 149 subjects	A significant reduction in IL-6 was seen following soft-tissue sleep surgery (SMD, 1.086; 95% CI, 1.952 to 0.221).
Yi et al., 2022 [99]	21 articles, 1047 patients	Upon therapy, the level of IL-6 significantly decreased—WMD −3.21.
Ning et al., 2019 [92]	6 RCTs, 546 subjects	The overall MD was 0.15, *p* = 0.23, and no discernible decrease in IL-6 levels was seen after CPAP therapy. Moderate heterogeneity was detected.
Zhong et al., 2015 [101]	20 studies, 646 patients	No significant result was observed for IL-6 serum levels after CPAP treatment (SMD = −0.24, 95% CI = −0.73 to 0.26).
Xie et al., 2013 [95]	16 studies, 491 subjects	According to several studies, IL-6 is the best predictor of changes in AHI and had an independent impact on OSA, although analysis produced inconclusive findings for IL-6 (SMD = 0.299 (95% CI, 0.001–0.596)).

Abbreviations: IL-6, interleukin-6; OSA, obstructive sleep apnea; BMI, body mass index; AHI, apnea–hypopnea index; CPAP, continuous positive airway pressure.

**Table 5 jcm-12-00060-t005:** Meta-analyses of TNA-α studies.

Authors	Material	Conclusion
Comparison to the control group		
Yi et al., 2022 [89]	34 studies, 1981 OSA patients and 1112 controls in	TNF-α levels were raised in OSA, and these rising tendencies were severity-dependent. Mendelian randomization did not reveal a causal link between OSA and increased TNF-α (WMD (95% CI)= 5.86 (4.80–6.93) pg/mL, *p* < 0.00001).
Imani et al., 2020 [103]	29 studies, 2718 OSA patients, 1893 controls (serum TNF-α) 17 studies, 1021 OSA patients, 601 controls (plasma TNF-α)	Adult patients with OSA had considerably higher plasma (pooled MD 5.90 pg/mL (95% CI = 4.00, 7.80; *p* < 0.00001) and serum (pooled MD 10.22 pg/mL (95% CI = 8.86, 11.58; *p* < 0.00001) TNF-α levels than controls. Increased TNF-α levels in OSA patients seemed to correlate with the severity of the condition.
Cao et al., 2020 [104]	50 articles 3503 OSA patients and 3379 healthy controls	Patients with OSA had TNF-α levels that were 1.77 (95% CI, 1.37 to 2.17, I^2^ = 97.8%, *p* < 0.0001) times greater than those in the control group. The concentration of TNF-α positively correlated with the severity of OSA. Older age substantially predicted a larger impact size of TNF-α level in OSA patients, according to meta-regression.
Li et al., 2017 [105]	59 studies, 2857 OSA patients and 2115	Circulating TNF-αlpha was considerably greater in patients than in controls (WMD]: 9.66 pg/mL, 95% confidence interval (CI): 8.66 to 11.24, *p* < 0.001), suggesting that TNF-αlpha may be a viable circulating biomarker for OSA development.
Nadeem et al., 2013 [30]	19 studies with 1316 subjects pooled for TNF-α	TNF-α was higher in patients with OSA than in control individuals (pooled SMD 1.03). The moderate trend of the age, BMI, and AHI’s substantial effect was also seen in meta-regression; the level of TNF-α was correlated with OSA severity.
**Change after treatment**		
Wang et al., 2022 [90]	15 studies, 454 patients	After using CPAP for less than three months, there was a slight drop in TNF-α. The total pooled SMD for TNF-α was 0.49.
Lee et al., 2022 [102]	8 studies, 205 subjects	Elevated TNF-α can be decreased by successful sleep surgery (SMD −0.56 (95% CI, −0.85 to −0.27).
Yeo et al., 2022 [88]	6 studies, 173 patients	TNF-α can be decreased by successful soft-tissue sleep surgery (SMD = −0.822).
Yi et al., 2022 [89]	4 articles, 125 patients, 109 controls	TNF-α was considerably reduced following CPAP management as compared to the non-CPAP group (WMD = 4.44 pg/mL, *p* < 0.00001).
Ning et al., 2019 [92]	3 RCTs, 311 patients	The treatment effect of CPAP for TNF-α was not statistically significant (MD 0.61; *p* = 0.35) with indications of considerable heterogeneity.
Xie et al., 2013 [95]	12 studies, 403 patients	TNF-α levels were found to decrease significantly (SMD, 0.478, *p* = 0.000). Subgroup analysis revealed that TNF-α levels decreased after less than three months of PAP treatment.
Baessler et al., 2013 [94]	8 studies, 165 patients	TNF-α and inflammatory marker levels in the blood are considerably reduced in OSA patients who use CPAP (pooled MD 1.14; CI (95%) 0.12 to 2.15, *p* = 0.03).

Abbreviations: TNF-α, tumor necrosis factor-alpha; BMI, body mass index; AHI, apnea–hypopnea index; OSA, obstructive sleep apnea syndrome; CPAP, continuous positive airway pressure.

**Table 6 jcm-12-00060-t006:** Meta-analyses of IL-8 studies in OSA adults.

Authors	Material	Conclusion
Comparison to the control group		
Li et al., 2021 [106]	25 studies, 2301 patients, 1123 controls	In patients with OSA, levels of IL-8 are elevated. The severity of OSA has a beneficial impact on this effect; the higher the AHI, the higher the levels.
Nadeem et al., 2013 [30]	16 studies (serum IL-8) 4 studies (plasma IL-8)	Patients with OSA had significantly elevated serum IL-8 concentrations compared with controls (SMD = 0.997, 95% CI = 0.437–1.517, *p* < 0.001). Individuals with OSA had significantly elevated plasma IL-8 concentrations compared with controls (pooled SMD 4.22).
Yi et al., 2022 [99]	13 studies, 1267 patients, 574 controls	Patients with OSA had higher serum concentrations of IL-8 than controls, although the difference was not statistically significant.
**Change after treatment**		
Xie et al., 2013 [95]	3 studies, 101 patients	IL-8 levels decreased after OSA treatment (SMD, 0.645 (95% CI, 0.362–0.929); z = 4.46; *p* = 0.000).

Abbreviations: OSA, obstructive sleep apnea; IL-8, interleukin-8; AHI, apnea–hypopnea index.

**Table 7 jcm-12-00060-t007:** Meta-analyses of ICAM-1 and VCAM-1 studies in OSA adults.

Authors	Material	Conclusion
Comparison to the control group		
Nadeem et al., 2013 [30] De Luca Canto et al., 2015 [97]	51 studies, 2952 patients, 2784 controls 8 studies, 495 participants (serum ICAM-1) 6 studies, 269 participants (serum VCAM-1) 1 article, 39 patients, 34 controls—Diagnostic capability-focused study	Significantly higher serum ICAM-1 level in subjects with OSA (pooled SMD 2.93). Significantly higher serum VCAM-1 level in subjects with OSA (pooled SMD 2.08). Potential biomarkers to distinguish OSA from non-OSA adults include ICAM-1 and VCAM-1, although with low diagnostic values. ICAM-1′s sensitivity/specificity (%) was 69/82, whereas VCAM-1′s was 74/65.
**Change after treatment**		
Tian et al., 2021 [107]		ICAM-1 was significantly reduced by CPAP therapy (SMD = 0.283, 95% CI 0.464 to 0.101, *p* = 0.002). However, after receiving CPAP therapy, VCAM-1 levels did not significantly change (SMD = 0.160, 95% CI: 0.641 to 0.320, *p* = 0.513).

Abbreviations: ICAM-1, intercellular adhesion molecule 1; VCAM-1, vascular cell adhesion molecule 1; OSA, obstructive sleep apnea; CPAP, continuous positive airway pressure therapy; SMD, standardized mean difference.

**Table 8 jcm-12-00060-t008:** Meta-analyses of VEGF studies in OSA adults.

Authors	Material	Conclusion
Comparison to the control group		
Qiu et al., 2020 [108]	5 relevant studies involving 262 patients	Patients with OSA had higher levels of VEGF than controls (SMD = 0.37, 95% CI = 0.90 to 1.63, *p* = 0.000).
Zhang et al., 2016 [109]	15 studies, 426 patients, 271 controls	The VEGF in the OSA group is relatively higher than that in the control group (SMD 1.89, 95% CI 0.92–2.87, *p* = 0.000). In subjects with age ≥50 years, higher VEGF concentrations were identified in the OSA group when compared with the control group, but they failed to find any difference in VEGF between the two groups in subjects aged <50 years. The increased VEGF levels were observed in OSA patients compared to controls when using serum to detect VEGF, whereas the VEGF levels did not differ between the two groups when using plasma for VEGF detection.
**Change after treatment**		
Qi et al., 2018 [110]	3 studies, 392 patients	Before and following CPAP therapy, VEGF levels in patients with OSA significantly decreased (SMD = −0.440, 95% (CI) = −0.684 to −0.196, z = 3.53, *p* = 0.000).

Abbreviations: VEGF, vascular endothelial growth factor; OSA, obstructive sleep apnea; CPAP, continuous positive airway pressure.

**Table 9 jcm-12-00060-t009:** Meta-analyses of homocysteine studies in OSA adults.

Authors	Material	Conclusion
Comparison to the control group		
Li et al., 2013 [111]	10 studies, 457 cases, 316 controls	Patients with OSA had a higher serum Hcy level than healthy controls (Hcy levels were 2.40 μmol/L higher in OSA patients than in control group (95% CI: 0.6 to 4.20, *p* < 0.01). In addition, this difference is more significant in moderate or severe OSA patients.
Niu et al., 2014 [112]	10 studies, 413 patients, 344 controls	Homocysteine levels were found to be 3.11 mmol/L higher in OSA patients compared to control subjects (95% CI: 2.08 to 4.15, *p* < 0.01).
**Change after treatment**		
Chen et al., 2014 [113]	6 studies, 206 cases	CPAP caused a statistically significant decrease in Hcy levels (WMD −0.62 (95% (CI) −1.21 to −0.04, *p* < 0.05).

Abbreviations: OSA, obstructive sleep apnea; Hcy, homocysteine; CPAP, continuous positive airway pressure.

**Table 10 jcm-12-00060-t010:** Meta-analyses of MDA studies in OSA adults.

Authors	Material	Conclusion
Comparison to the control group		
Fadaei et al., 2020 [114]	14 studies, 867 cases, 429 controls	MDA serum concentration significantly increased in OSA patients compared to the controls (SMD (95% CI): 1.18 (0.68, 1.68), *p* < 0.0001).
**Change after treatment**		
Chen et al., 2020 [115] Fadaei et al., 2020 [116]	10 studies, 220 cases 13 studies, 606 cases	A significant decrease in serum or plasma MDA was observed after CPAP treatment (SMD = 1.164, 95% CI = 0.443 to 1.885, z = 3.16, *p* = 0.002). CPAP therapy significantly reduced MDA levels in OSA patients (SMD −1.51 (95% CI, −2.06 to 0.97), *p* < 0.05).

Abbreviations: MDA, malondialdehyde; OSA, obstructive sleep apnea; CPAP, continuous positive airway pressure.

**Table 11 jcm-12-00060-t011:** Meta-analyses of leptin studies in OSA adults.

Authors	Material	Conclusion
Comparison to the control group		
Li et al., 2021 [117]	34 studies, 1485 cases, 1201 controls	Plasma leptin levels in adults with OSA were significantly higher than the corresponding levels in the control group (WMD = 3.80 ng/mL, 95% CI = 3.09–4.50, *p* < 0.00001).
**Change after treatment**		
Zhang et al., 2014 [118] Chen et al., 2015 [119]	15 studies, 427 cases 11 studies, 413 cases	A significant decrease in serum leptin levels was observed among patients with OSA after CPAP therapy (SMD = 0.137; 95% (CI) 0.002 to 0.272; *p* = 0.046). CPAP therapy significantly reduced leptin levels in OSA patients (WMD = 1.44; CI (95%): 1.11–1.77, *p* < 0.01).

Abbreviations: OSA, obstructive sleep apnea; CPAP, continuous positive airway pressure.

**Table 12 jcm-12-00060-t012:** Meta-analyses of IGF-1 studies.

Authors	Material	Conclusion
Comparison to the control group		
He et al., 2022 [121]	34 studies, 1407 cases, 1039 controls	IGF-1 levels in patients with OSA were significantly lower than those of healthy controls (SMD = −1.37, 95% CI = −1.78–0.96, *p* < 0.001).
**Change after treatment**		
Chen et al., 2014 [120]	9 studies, 168 cases	CPAP was associated with a statistically significant increase in IGF-1 in OSA patients (SMD = −0.436, 95% CI = −0.653 to −0.218, *p* = 0.000).

Abbreviations: IGF-1, insulin-like growth factor; OSA, obstructive sleep apnea; CPAP, continuous positive airway pressure.

**Table 13 jcm-12-00060-t013:** Meta-analyses of ALT and AST studies in OSA adults.

Authors	Material	Conclusion
Comparison to the control group		
Sookoian et al., 2013 [122]	11 studies, 668 cases, 404 controls	The SMD values of ALT levels were significantly different in OSA patients compared to controls. AST levels were significantly different in OSA patients compared to controls. There was an increase of 13.3% in ALT and 4.4% in AST levels in OSA patients.
**Change after treatment**		
Chen et al., 2016 [123]	5 studies, 192 cases	CPAP was associated with a statistically significant decrease in ALT in OSA patients (WMD = 8.036, 95% CI = 2.788 to 13.285, z = 3.00, *p* = 0.003). A significant difference in AST was observed before and after CPAP treatment (WMD = 4.612, 95% CI = 0.817 to 8.407, z = 2.38, *p* = 0.017).

Abbreviations: ALT, alanine transaminase; OSA, obstructive sleep apnea, AST, aspartate aminotransferase; CPAP, continuous positive airway pressure.

**Table 14 jcm-12-00060-t014:** Meta-analyses of lipid profile studies in OSA adults.

Authors	Material	Conclusion
Comparison to the control group		
Nadeem et al., 2014 [124]	64 studies, 18,116 patients (TC: 63 studies, 18,111 subjects; LDL 50 studies, 13,894 subjects; HDL: 64 studies, 18,116 subjects; TG 62 studies, 17,831 subjects)	Patients with OSA have increased dyslipidemia (LDL, HDL, TG, TC). Pooled SDM for total cholesterol = 0.267 (*p* = 0.001); pooled SDM for LDL cholesterol = 0.296 (*p* = 0.001). Pooled SDM for HDL cholesterol = −0.433 (*p* = 0.001). Pooled SDM for the triglyceride = 0.603 (*p* = 0.001). Age has a considerable impact on TC, LDL, and HDL, according to meta-regression for age, BMI, and AHI. While AHI had a substantial impact on LDL and TG, BMI had a large impact on LDL and HDL.
**Change after treatment**		
Lee et al., 2022 [125]	13 studies, 710 adults	In subjects with OSA, sleep surgery may enhance the lipid profile. Triglycerides were reduced by 14.0 mg/dL (mean−14.0 mg/dL; 95% CI −22.2 to −5.8), LDL was reduced by 7.2 mg/dL (mean −7.2 mg/dL; 95% CI −11.0 to −3.3), and total cholesterol reduced by 7.7 mg/dL (mean −7.7 mg/dL; 95% CI −12.2 to −3.2). Improvements in lipid profile measures positively correlated with the degree of OSA improvement.
Wang et al., 2022 [90]	TC: 28 studies, 918 subjects; LDL 21 studies, 670 subjects; HDL: 26 studies, 864 subjects; TG 30 studies, 1021 subjects	PAP therapy lowers blood cholesterol levels in OSA patients. Short-term therapy might lower LDL levels in the blood (SMD: 0.40 (95% CI: 0.18–0.62); *p* = 0.000), but medium- or long-term CPAP treatment raised HDL levels (SMD: –0.20 (95% CI: –0.36 to –0.03); *p* = 0.018) and lowered TC levels (SMD: 0.20 (95% CI: 0.05–0.34); *p* = 0.007) in the blood.
Nadeem et al., 2015 [126]	29 studies, 1958 subjects TC: 24 studies, 1929 subjects; LDL 18 studies, 676 subjects; HDL: 23 studies, 806 subjects; TG 25 studies, 1926 subjects	Dyslipidemia that results from OSA therapy with CPAP appears to be improved (lower total and LDL cholesterol and higher HDL cholesterol). It does not seem to have an impact on TG levels. SMD for TC was −41.5 to −0.077, and PMD was -5.660 (*p* < 0.001). In LDL, SMD −3.7 to 0; PMD −0.488 (*p* < 0.001). For HDL, SMD −0.498 to 1.94; PMD was 0.207 (*p* < 0.01). For TG, SMD −9.327 to 1.98; PMD was −0.054 (*p* < 0.129).
Xu et al., 2014 [127]	6 RCTs, TC: 370 patients vs. 371 controls, LDL 276 patients vs. 274 controls, HDL 269 patients and 266 controls, TG 330 patients vs. 328 controls	The difference in mean TC levels between the study and control groups shows that CPAP therapy of OSA patients reduced metabolic dyslipidemia; TG, LDL, and HDL levels did not change. The difference in mean TC level between the CPAP and sham CPAP/control groups was statistically different according to the pooled estimate (0.15, *p* = 0.01).

Abbreviations: OSA, obstructive sleep apnea.

**Table 15 jcm-12-00060-t015:** Meta-analyses of adiponectin studies.

Authors	Material	Conclusion
Comparison to the control group		
Lu et al., 2019 [128]	20 studies, 1356 cases	The serum/plasma adiponectin levels were considerably lower in OSA patients than that in control subjects. It suggests a possible role of adiponectin in OSAHS pathogenesis. (SMD = −0.71, 95% CI = −0.92 to −0.49, *p* < 0.001).
**Change after treatment**		
Chen et al., 2015 [129]	11 studies, 240 cases (10 observational studies, 1 randomized controlled study)	Adiponectin levels in OSA patients did not differ between those taken before and after CPAP therapy (SMD = 0.059, 95% confidence interval (CI) = −0.250 to 0.368, z = 0.37, *p* = 0.710).
Iftikhar et al., 2015 [130]	3 RCTs with 200 participants (101 in CPAP and 99 in sham CPAP groups)	There was no significant change in adiponectin levels before and after CPAP treatment in OSA patients (SMD =−0.06 (95% CI: −0.28 to 0.15), *p* = 0.56).
Hecht et al. [131]	2 studies, 144 subjects	A significant change in adiponectin levels before and after CPAP treatment could not be observed.

Abbreviations: OSA, obstructive sleep apnea; CPAP, continuous positive airway pressure.

**Table 16 jcm-12-00060-t016:** Meta-analyses of cortisol studies.

Authors	Material	Conclusion
Comparison to the control group		
Imani et al., 2021 [132]	Serum (5 studies, 314 subjects)’ Plasma (6 studies, 253 subjects)	Cortisol concentrations in the serum and plasma did not differ between adults with OSA and healthy controls.
**Change after treatment**		
Ken-Dror et al., 2021 [133]	Plasma (15 prospective control studies, 335 subjects; 3 randomized control trials, 195 subjects)	CPAP treatment reduced plasma cortisol levels in prospective cohort studies: (SMD = −0.28) and in randomized control trials (SMD = −0.39).

Abbreviations: OSA, obstructive sleep apnea; CPAP, continuous positive airway pressure; SMD, standardized mean difference.

## Data Availability

The data that support the findings of this study are available from the corresponding author, upon reasonable request.

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
