# Peer review of "Potential Diagnostic and Monitoring Biomarkers of Obstructive Sleep Apnea–Umbrella Review of Meta-Analyses"

_jcm, 2022, doi:10.3390/jcm12010060_

Round 1

Reviewer 1 Report

I congratulate you on a very complex meta analisis regarding biomarkers in OSA cases.

Please verify the reference nr 28 and cite it properly.

Hope to see future work from your team also.

Author Response

RESPONSE TO REVIEWER 1:

The authors appreciate the thorough review and constructive comment. Our response and the changes to the manuscript after the Reviewers’ suggestions are written in red.

We have provided a response to Reviewer 1:

Reviewer 1:

I congratulate you on a very complex meta-analysis regarding biomarkers in OSA cases.

Please verify the reference nr 28 and cite it properly.

Hope to see future work from your team also.

We thank the Reviewer for this comment. We have verified reference 28 and changed the citation. After changes to the manuscript, it can be found under the number 81.

Reviewer 2 Report

I thank the editor for sending this interesting article. My comments are:

1. It is not necessary to display the AMSTAR instrument. If you still want to include it, please leave it in supplementary material.

2. Table 3 is too long and cumbersome to read. I suggest sending to supplementary material and only presenting a general summary of the items included in AMSTAR under a subheading called "quality of studies"

3. Since it is an umbrella review, why were specific search engines for systematic reviews such as epistemonikos not included?

4. At the beginning of each biomarker in the results section, there is an explanation of each of them. This section should be limited only to presenting the results and not to showing the theoretical background of the outcomes evaluated. That's what the introduction is for.

5. A problem that systematic reviews with meta-analyses usually present is the overlapping of primary studies, which causes results to be overrepresented. Because study overlap was not assessed (https://pubmed.ncbi.nlm.nih.gov/352780)

Author Response

RESPONSE TO REVIEWER 2:

The authors appreciate the thorough review and constructive comments. We have carefully assessed each of the comments and criticisms and we made changes in the manuscript accordingly. We believe that the revised manuscript has improved through this review process. We hope that the Editor and the Reviewers will agree with us. All changes to the manuscript after the Reviewer's suggestions are written in red.

We have provided a detailed response to Reviewer 2:

  1. It is not necessary to display the AMSTAR instrument. If you still want to include it, please leave it in supplementary material.

We thank the Reviewer for this comment. We have moved table 2 to the supplementary material.

  1. Table 3 is too long and cumbersome to read. I suggest sending to supplementary material and only presenting a general summary of the items included in AMSTAR under a subheading called "quality of studies"

We appreciate this suggestion. The length of the AMSTAR quality assessment is due to a large number of meta-analyses included and graded in our review. As proposed, we moved Table 3 to the supplementary material and added a paragraph presenting a general summary of the results. 

  1. Since it is an umbrella review, why were specific search engines for systematic reviews such as epistemonikos not included?

We agree with the Reviewer that Epistemonikos is a comprehensive database of systematic receives. However, we decided to make a search using PUBMED as our main source and an additional Elsevier database (SCOPUS) library in the methodology of our manuscript. The two databases chosen by us are similar, if not better, in the sensitivity and specificity in retrieving valuable research in comparison to Epistemonikos. 

(vide Rathbone J, Carter M, Hoffmann T, Glasziou P. A comparison of the performance of seven key bibliographic databases in identifying all relevant systematic reviews of interventions for hypertension. Syst Rev. 2016 Feb 9;5:27. doi: 10.1186/s13643-016-0197-5. PMID: 26862061; PMCID: PMC4748526.)

It would be valuable to see the systematic review of meta-analyses retrieved from more databases, including Epistemonikos, Cochrane, or TRIP. 

We have added a section to the limitations of the study about the search strategy in our review.

  1. At the beginning of each biomarker in the results section, there is an explanation of each of them. This section should be limited only to presenting the results and not to showing the theoretical background of the outcomes evaluated. That's what the introduction is for.

            OSAS is associated with an increase in serum and plasma concentrations of dozens of compounds compared to the non-OSAS control groups. Before conducting the review, we couldn’t have known which potential biomarkers meet the inclusion conditions, resulting in the explanation for each potential biomarker written in the results section. 

As per the Reviewer’s suggestion, we have moved the explanation section to the introduction.

  1. A problem that systematic reviews with meta-analyses usually present is the overlapping of primary studies, which causes results to be overrepresented. Because study overlap was not assessed (https://pubmed.ncbi.nlm.nih.gov/352780)

This systematic review of the existing meta-analyses did not intend to combine data from the meta-analyses and do a combined statistical analysis using the meta-analysis data. If this was the case, the original study overlap in different meta-analyses would result in an error. If that was the intention, the dissection of each meta-analysis and selection of original studies that are not represented in any other meta-analysis would not only require re-conduct a new meta-analysis of each new set of an original set of studies (except those that are removed), would pose the challenge to decide to include which original study in which meta-analysis (while eliminating from the others). Not only does no outside (other than the original authors of a meta-analysis) investigator have any authority to revise any existing meta-analysis by removing some of the studies from that, but it would be impossible to conduct such an analysis. The authors of this manuscript do not claim to do a meta-analysis of the selected meta-analyses, but rather have done a review of the meta-analyses that summarize and evaluate the quality of their material, methods, and results. Therefore, our manuscript does not have the problem of overlapping studies that are included in different meta-analyses. On the other hand, this issue may need better clarification in the manuscript.  Not under the "weaknesses" section, but rather the discussion that clarifies what this manuscript is about, and what it is not. 

Once again, the authors would like to thank the Reviewer for the contribution to improving the manuscript. If any additional corrections or explanations are needed, we are ready to do them immediately.

Reviewer 3 Report

Dear

1. There are some grammatical errors.

2. Please pay attention to abbreviations. For example, you write Apnea Syndrome (OSA) several times in the text.

3. Please add the full name of ICAM-1, VCAM-1, ALT, and AST and lipid profile (TC, LDLc, HDLc, TG) for the first time.

4. The study missed several meta-analyses and should update your manuscript. Why did not the authors select

the rest of the factors such as adiponectin, cortisol, etc.?  10.3390/medicina58091266.

5.

Author Response

RESPONSE TO REVIEWER 3:

The authors appreciate the thorough review and constructive comments. We have carefully assessed each of the comments and criticisms and we made changes in the manuscript accordingly. We believe that the revised manuscript has improved through this review process. We hope that the Editor and the reviewers will agree with us.  All changes to the manuscript after the reviewer's suggestions are written in red.

We have provided a detailed response to Reviewer 3:

Reviewer 3:

  1. There are some grammatical errors.

We thank the Reviewer for this comment. Grammatical errors were corrected.

  1. Please pay attention to abbreviations. For example, you write Apnea Syndrome (OSA) several times in the text.

We double-checked the abbreviations to make sure they are correctly used in the whole manuscript. 

  1. Please add the full name of ICAM-1, VCAM-1, ALT, and AST and lipid profile (TC, LDLc, HDLc, TG) for the first time.

We provided explanations for the abbreviations. As a consequence of changes to the manuscript suggested by another reviewer, all the names of chosen compounds, and the basis of their pathophysiological connection to OSA, are moved to the introduction section. 

  1. The study missed several meta-analyses and should update your manuscript. Why did not the authors select the rest of the factors such as adiponectin, cortisol, etc.? 10.3390/medicina58101499. 10.3390/medicina58091266.

We highly appreciate this suggestion. Although the study selection was done by two authors independently, the omission was possible due to a large number of studies regarding OSA studies. We have added the missing studies regarding cortisol and adiponectin to our manuscript. 

Round 2

Reviewer 2 Report

The authors have adequately responded to all my comments.